

# Assessment of cloud related fine mode AOD enhancements based on AERONET SDA product

A. Arola[1], T.F. Eck[2,3], H. Kokkola[1], M. R. A. Pitkänen[1,4], and S. Romakkaniemi[1]

[1]Finnish Meteorological Institute, Kuopio, Finland.
[2]Universities Space Research Association, Columbia, MD, USA.
[3]NASA Goddard Space Flight Center, Greenbelt, MD, USA.
[4]Department of Applied Physics, University of Eastern Finland, Kuopio.

*Correspondence to:* Antti Arola
(antti.arola@fmi.fi)

**Abstract.**

AERONET (AErosol RObotic NETwork), which is a network of ground-based sun photometers, produces a data product called the Aerosol Spectral Deconvolution Algorithm (SDA) that utilizes spectral total extinction AOD data to infer the component fine and coarse mode optical depths at 500nm. Based on its assumptions, SDA identifies cloud optical depth as the coarse mode AOD component and therefore effectively computes the fine mode AOD also in mixed cloud-aerosol observations. Therefore, it can be argued that the more representative AOD for fine mode fraction should be based on all direct sun measurements and not only on those cloud-screened for clear-sky conditions, in other words on those from Level 1 (L1) instead of Level 2 (L2). The objective of our study was to assess, including all the available AERONET sites, the magnitude of this cloud enhancement in fine mode AOD, in other words contrasting SDA L1 and L2 in our analysis. Assuming that the cloud-screening correctly separates the cloudy and clear-sky conditions, then the increases in fine mode AOD in can be due to various cloud-related processes, mainly by in-cloud processing, hygroscopic growth and new particle formation from gas-to-particle conversion in aqueous phase in cloud droplets. We estimated these cloud-related enhancements in fine mode AOD seasonally and found, for instance, than in June-August season the average over all the AERONET sites was 0.011, when total fine mode AOD from L2 data was 0.154, therefore the relative enhancement was 7%. The enhancements were largest, both absolutely and relatively, in East-Asia; for example in June-August season the absolute and relative differences in fine mode AOD, between L1 and L2 measurements, were 0.022 and 10%, respectively. Corresponding values in North-America and Europe were about 0.01 and 6-7%. In some some highly polluted cities the enhancement is greater than these regional averages, e.g. in Beijing and in JJA season the corresponding absolute values were about 0.1. It is



difficult to separate the fine mode AOD enhancements due to in-cloud processing and hygroscopic growth, but we attempted to get some understanding by conducting a similar analysis for SDA-based

fine mode Angstrom Exponent (AE) patterns. Moreover, we exploited a cloud parcel model, in order to understand in more depth the relative role of the processes inducing the changes in the effective fine mode particle size, and thus the changes in fine mode AE.

## 1   Introduction

Aerosol–cloud interactions contribute the largest uncertainty to the total anthropogenic radiative

forcing (Myhre, 2013). One of the issues that hinder the measurement-based assessment of aerosol-cloud interactions by remote sensing methods is that typically aerosols and clouds cannot be measured simultaneously by passive remote sensing methods, including ground-based sun-photometers. In these techniques, so-called cloud-screening algorithms are therefore applied to provide aerosol optical depth (AOD) measurements for clear-sky conditions only. Due to this limitation, in aerosol-

cloud interaction studies, aerosol and cloud properties have inherently different temporal sampling and therefore additional effects, e.g. impact of meteorology, have necessarily a possible influence in the derived correlations.

Many observational studies have found positive correlations between cloud fraction and AOD (Ignatov et al., 2005; Chand et al., 2012)). However, as stressed above, with passive remote sensing

the AOD measurements in cloudy conditions are not possible and thus these studies have to rely on cloud-screening technique and therefore the derived cloud aerosol relationships might be linked more to cloud contamination than to real physical processes. On the other hand, active remote sensing of aerosol from lidar measurements does not suffer similarly from this issue of cloud adjacency and these data have been analyzed as well for cloud aerosol interaction effects. Cloud–Aerosol Lidar and

Infrared Pathfinder Satellite Observations (CALIPSO) lidar data over oceans have been investigated for the relationship between aerosol and clouds (e.g., Várnai and Marshak, 2011; Yang, 2015). These studies have shown a sharp increase in aerosol signal within 4 km towards clouds

The physical mechanisms contributing to the positive correlation between AOD and cloudiness, in addition to unphysical contamination by undetected clouds, are mainly the following: hygro-

scopic growth of aerosol particles, meteorological conditions, and in-cloud processing (e.g. due to the aqueous process including nitrate or sulfate). These effects, particularly hygroscopic growth versus meteorological influence, have been debated (e.g., Mauger and Norris, 2007; Engström and Ekman, 2010). However, it is a challenging task to separate the influence of each factor and thus, they have remained poorly known. One of the challenges is related to the time scale of physical

processes involved; new particle formation as gas-to-particle processes occur in minutes in cloud droplets compared to days in cloudless air (i.e. sulfate formation as an example).





AERONET (AErosol RObotic NETwork), which is a network of ground-based sun photometers, includes also so-called Aerosol Spectral Deconvolution Algorithm (SDA) that utilizes spectral total extinction AOD data to infer the component fine and coarse mode optical depths at 500nm. Based on its assumptions, SDA identifies cloud optical depth as the coarse mode AOD component and therefore effectively computes the fine mode AOD also in mixed cloud-aerosol observations. Therefore, these measurements provide interesting insight into the simultaneous aerosol cloud measurements. More specifically, one can obtain and separate aerosol information in clear-sky and cloudy sky conditions, when clouds are thin enough that the direct sun measurements are possible.

AERONET SDA product has been used to some extent, i.e. for rapid AOD increases in the vicinity of cumulus (Eck et al., 2014); but nevertheless it has not been fully exploited yet and thus its unique features offer potential for additional interesting studies. In this paper, we present an analysis of cloud enhanced AOD measurements, based on AERONET SDA product, including all the AERONET sites.

## 2 Data and Methods

### 2.1 AERONET data

AERONET (AErosol RObotic NETwork) is a globally distributed network of automatic sun and sky scanning radiometers that measure at several wavelengths, typically centered at 0.34, 0.38, 0.44, 0.50, 0.67, 0.87, 0.94, and 1.02 $\mu$m. Each band has a full width of approximately 0.010 $\mu$m at half maximum (FWHM), except for 0.34 $\mu$m and 0.38$\mu$m channels that have FWHM of 0.002 $\mu$m. All of these spectral bands are utilized in the direct Sun measurements, while four of them are used for the sky radiance measurements, 0.44, 0.67, 0.87 and 1.02 $\mu$m. Spectral aerosol optical depth (AOD) is obtained from direct sun measurements at high accuracy ( 0.01 to 0.02 for overhead sun, with the larger errors in the UV (Eck et al., 1999)). The inversion product includes other aerosol optical properties, such as single scattering albedo (SSA), refractive indices and the column integrated aerosol size distributions above the measurement site provided at the sky radiance wavelengths (Holben et al., 1998; Dubovik et al., 2000).

The spectral deconvolution algorithm (SDA) product, and its ability to separate coarse and fine mode AOD and provide useful fine mode AOD also in cloudy conditions, is vitally important in our study. O'Neill et al. (2001, 2002) developed SDA algorithm that utilizes spectral total extinction AOD data, with the assumption of bimodal aerosol size distributions, to infer the component fine and coarse mode optical depths. An additional fundamental assumption of the algorithm is that the coarse mode Ångström exponent (AE) and its derivative are assumed to be -0.15 and zero, respectively. The Ångström exponent and its spectral variation of measured total AOD (dAE/dlnWL) are the measurement inputs to the algorithm. These are determined from spectral AOD measurements at 5 wavelength: 380, 440, 500, 675, and 870 nm. In order to assume good quality SDA retrievals we



required these 5 wavelength be available in the Level 2 data and the L1 data were only utilized when L2 data were available in the same week. Additionally, a consistency check of measured AOD compared to SDA retrieved total AOD at 500 nm was applied to both the L1 and L2 SDA data.

The strength of this algorithm is that fine mode AOD can be obtained also in cloudy conditions, as demonstrated by O'Neill et al. (2002), since it identifies cloud optical depth as the coarse mode AOD component. Moreover, Chew et al. (2011), by comparing AERONET measured spectral AOD with lidar data, showed that SDA is able to effectively separate the fine and coarse so that the latter is only influenced by clouds. It is likely, however, that the fine mode AOD is underestimated, when cirrus

ice crystal clouds overlay the aerosol. This occurs due to strong forward scattering into the field-of-view of the sun-photometer (A. Smirnov: personal communication, 2016). However cloud screening also occurs when cirrus is not present (high temporal variance in the presence of clouds ; Eck et al. (2014)) and also when very few AOD observations occur on a primarily cloudy day. Nonetheless, since some of the cloud observations occur with cirrus present, the SDA overall provides a lower

limit on the enhancement of fine AOD in the presence of clouds.

In our analysis, we included fine mode AOD and AE at 500 nm, from both Level1 and Level2 SDA measurements, the latter for all-sky conditions and the former for clear-sky conditions. This data version (Version 2) includes cloud screening of Smirnov et al. (2000), which has been used for all papers using AERONET data since 2000 when this cloud screening algorithm was implemented.

New version 3 is very likely to be released sometime in 2016, with significantly different cloud screening.

Moreover, we constructed our own specific "Level0" data set of SDA measurements, including only those cases of Level1 that were not in Level2, thus these include only cases when it was cloudy, according to the cloud-screening. From these different data set, corresponding to the different cases

of cloudiness, we first calculated hourly means and averaged them to obtain the daily mean values. For the monthly averaged values, we required at least 10 days of data.

## 3   Results

### 3.1   Spatial and temporal patterns of cloud induced AOD and AE

We conducted our analysis first for all the available AERONET sites on a seasonal basis, for the fol-

lowing seasons: March-May (MAM), June-August (JJA), September-November (SON), December-February (DJF). Figures 1 to 4 shows these seasonal cases of the difference in AOD between Level0 and Level2 data, thus between cases of solely cloudy or clear-sky AOD measurements. When aerosol climatologies, monthly AOD means or other statistics are formed, then Level2 data of clear-sky measurements (excluding cloudy cases) are usually only included. This has to be definitely done with

passive satellite measurements, in order to avoid cloud contamination in total AOD. However, this can lead to systematic biases due to the sampling; clear-sky conditions are related to particular type




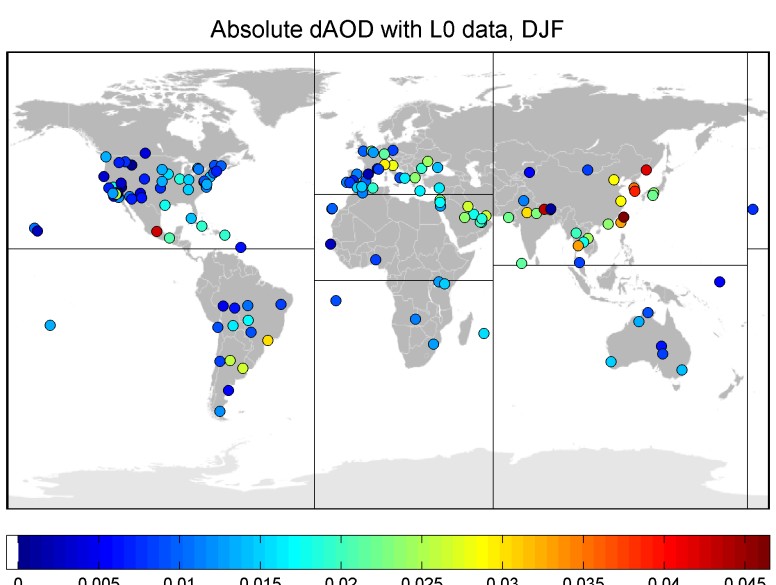

**Figure 1.** The difference between cloudy-sky and clear-sky fine mode AOD for DJF season.

of weather patterns, while the excluded cloudy cases may differ systematically also in their aerosol loading. Therefore, we want to stress that by our analysis, we can now obtain a quantitative estimate for the fine mode AOD that is more representative for all-sky conditions and thus also for the

enhancement due to these cloudy cases.

We additionally sub-divided our results into the following seven regions, shown also by lines in the plots: North-America, South-America, Europe, North-Africa, South-Africa, Asia, and Australasia. Table 1 shows the seasonal results for each region, e.g. the enhanced fine mode AOD, if sampled only for clear-sky conditions compared to cloudy-sky or all-sky cases. We can see that the AOD

enhancements are consistently largest throughout the year in the Asia, reaching values of about 0.1 in many sites. This is a substantial difference, that would not have a negligible effect in the radiative effect estimates either, if Level2 data were used instead. On the other hand, the difference over all the included sites is rather notable as well, e.g. in JJA fine mode AOD of all-sky data is 0.011 higher than the mean based on Level2 only (0.154), thus all-sky fine mode AOD being about 7% higher.





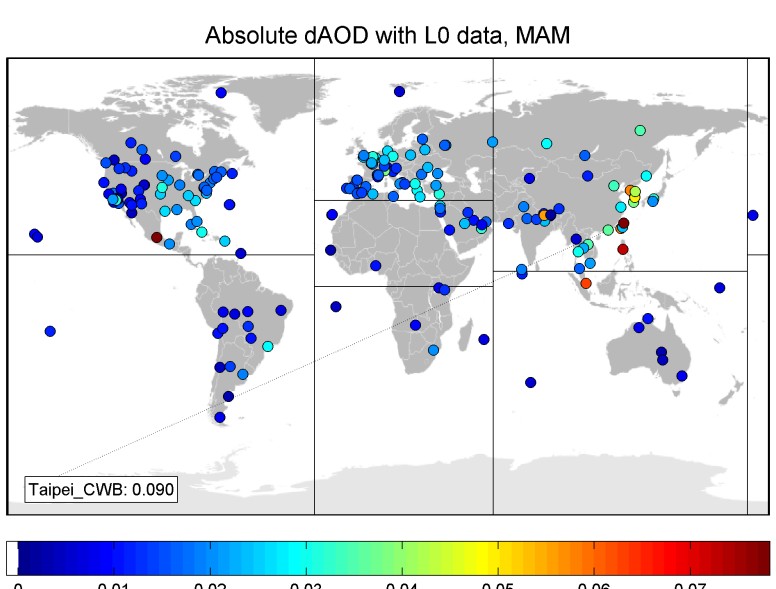

**Figure 2.** Otherwise similar to the Figure 1, but for MAM season.

We also established the AOD differences on a monthly basis separately for each AERONET site. Moreover, we made it additionally and similarly for fine mode AE parameter, which was weighted by AOD. We considered this AOD weighting both necessary and useful, in order to produce more robust signal for the seasonality; otherwise the within month AE variability was substantially higher, clearly due to the cases of lowest AOD when the AOD magnitude approaches the uncertainty of the single

channel AOD itself. Figures 5 to 14 show examples of this analysis for a selected set of stations. Figure 5 shows the weighted $dAE$ (difference in AE between Level0 and Level2 data) monthly seasonality for Arica, Chile, while Figure 6 shows the corresponding annual pattern for $dAOD$ (difference in AOD between Level0 and Level2). Similar pair of figures are shown for also for four other sites, in this order: Gwangju GIST, Korea; Taihu and XiangHe in China; Walker Branch, USA.

The patterns of $dAE$ do not show a strong seasonality. Moreover, generally the timing of highest $dAOD$ is typically related to slightly negative $dAE$, perhaps most evidently in XiangHe and Walker Branch. In other words, AE of cloudy cases is slightly smaller, suggesting somewhat larger particles, likely related to swelling in humid conditions. For cumulus clouds in the mid-Atlantic US the $dAE$




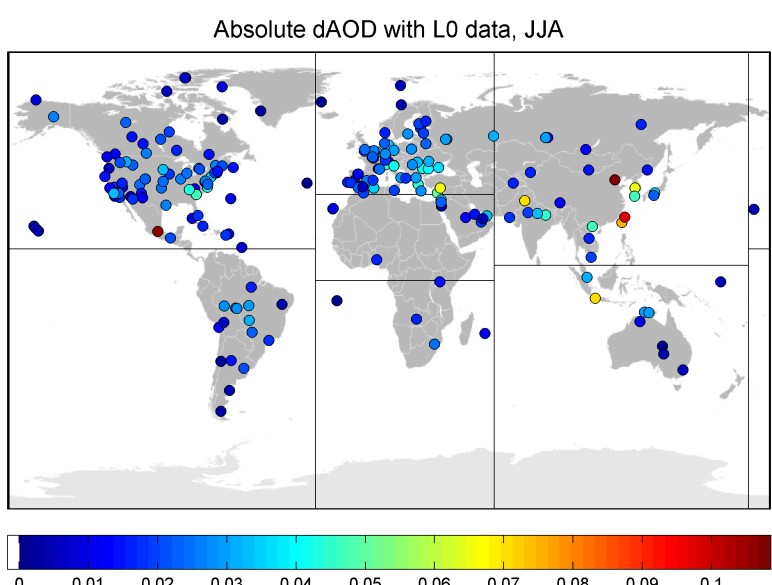

Absolute dAOD with L0 data, JJA

**Figure 3.** Otherwise similar to the Figure 1, but for JJA season.

did not change much despite large changes in AOD on some days. This strongly suggested that
particles grew in size from sub-visible Aitken, in addition to larger particle swelling in the high RH
environment in and near clouds.

In such a polluted environment (Baltimore-Washington region) where it is known that there is SO2
present it is also highly likely that sulfate particle formation also occurs in the clouds (rapid SO2
to sulfate conversion in aqueous phase versus relatively slow in non-cloudy environments). Other
gas-to-particle conversions are likely in the aqueous phase in cloud droplets (nitrates and organic
particles; (e.g., Hayden et al., 2008; Ervens et al., 2011)). Therefore near zero change in AE could
mean both processes are counter-balancing each other.

## 3.2 Cloud parcel model based investigation of cloud-induced AOD and AE changes.

The fine mode AE differences, between cloudy and clear-sky cases, shown in the previous section
exhibited typically negative values. And as discussed above, these cases are likely dominated by
particle growth in the humid conditions over the cloud activation. The former process results in




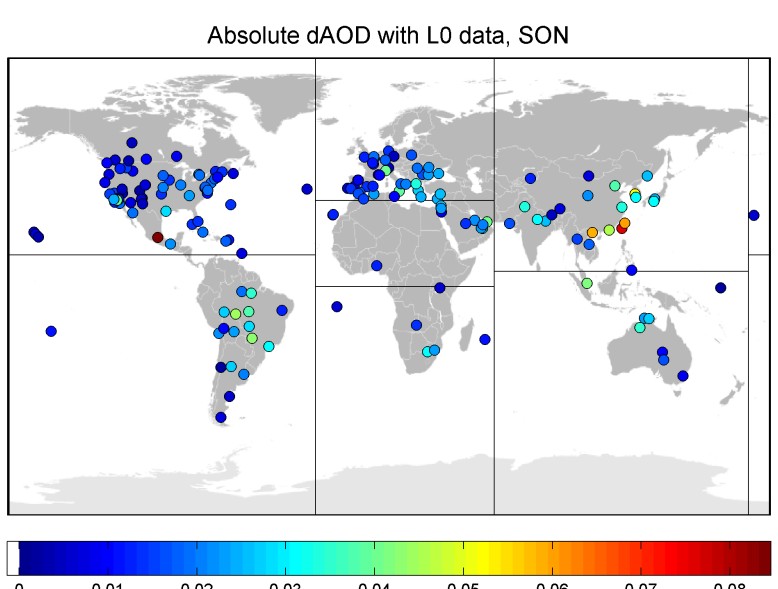

**Figure 4.** Otherwise similar to the Figure 1, but for SON season.

increase in the effective wet particle size. On the other hand, if the size threshold of droplet formation reached small enough particle sizes to affect the fine mode AOD, it would have a competing effect to decrease the effective size in the fine mode. In fact, there were only about ten sites of clearly

positive $dAE$, all being either Island or coastal sites, thus strongly affected by marine aerosols that typically has a strong bimodality with relatively small particles in the Aitken/accumulation mode, (e.g., Heintzenberg et al., 2004). Figure 15 shows, as an example, the weighted fine mode $dAE$ of Lanai, which is the sixth-largest of the Hawaiian Islands. Admittedly, these positive values are not very large. However, particularly during the summer the negative cases seem to be essentially

missing, and thus our main interest is to quantitatively understand the processes and their relative importance that result in prevailing positive fine mode $dAE$.

With the help of a cloud parcel model, we investigated in more detail the relative role of aerosol hygroscopic growth and cloud activation on the enhancement of AOD. This was done for different cloudy conditions, namely Lanai and Walker Branch that represent totally distinct aerosol condi-

tions. The cloud parcel model employed here has been used in several publications and has been





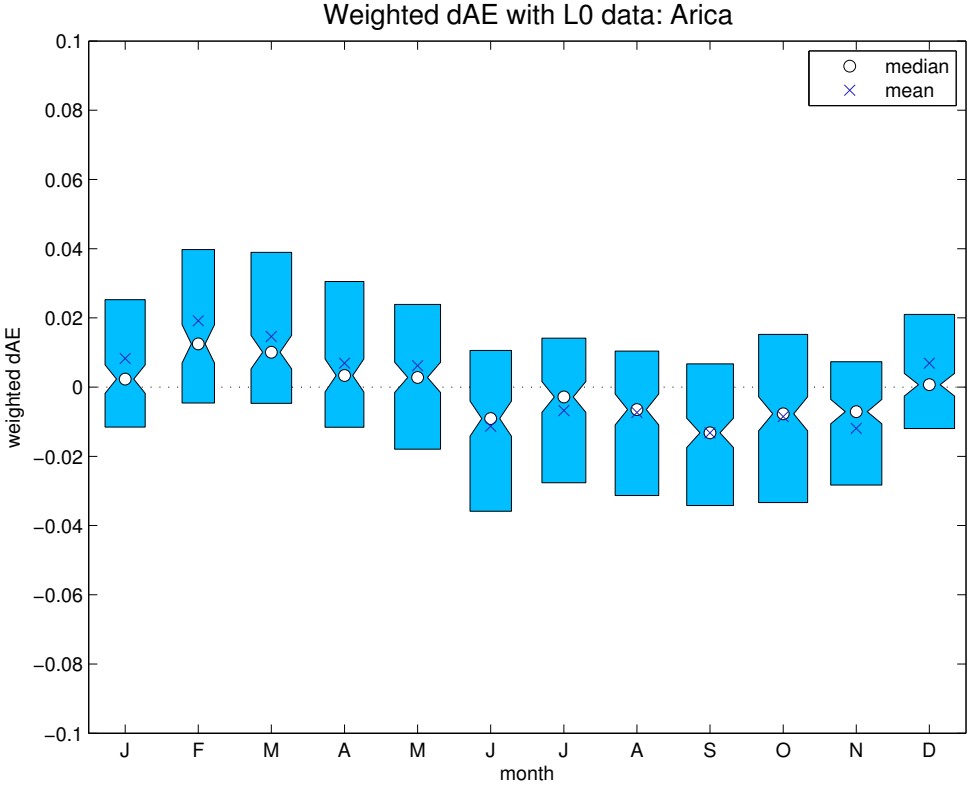

**Figure 5.** Monthly $dAE$ (difference in L0 AE and L2 AE) for Arica, Chile.

described in detail in Kokkola et al. (2003). In short, the model solves condensation and evaporation of water between the gas and particle phase. It has a sectional representation of aerosol particle size distribution with a detailed size dependent description of aerosol composition. The model can be employed to study how aerosol size distribution and composition are affecting the wet size of particles
and cloud droplet formation in different atmospheric conditions. Especially interesting here is the partitioning of aerosol particles between cloud droplets and interstitial aerosol, the latter contributing to fine mode AOD.

There are cloud related processes that our model does not fully describe, including gas-to-particle conversion and chemical reactions occurring in the aqueous phase. Also, the particle growth in hu-
mid regions in surrounding clouds is not included in our 1-D exercise. However, arguably the most important processes of marine cloud environment are included by our cloud parcel modeling study.





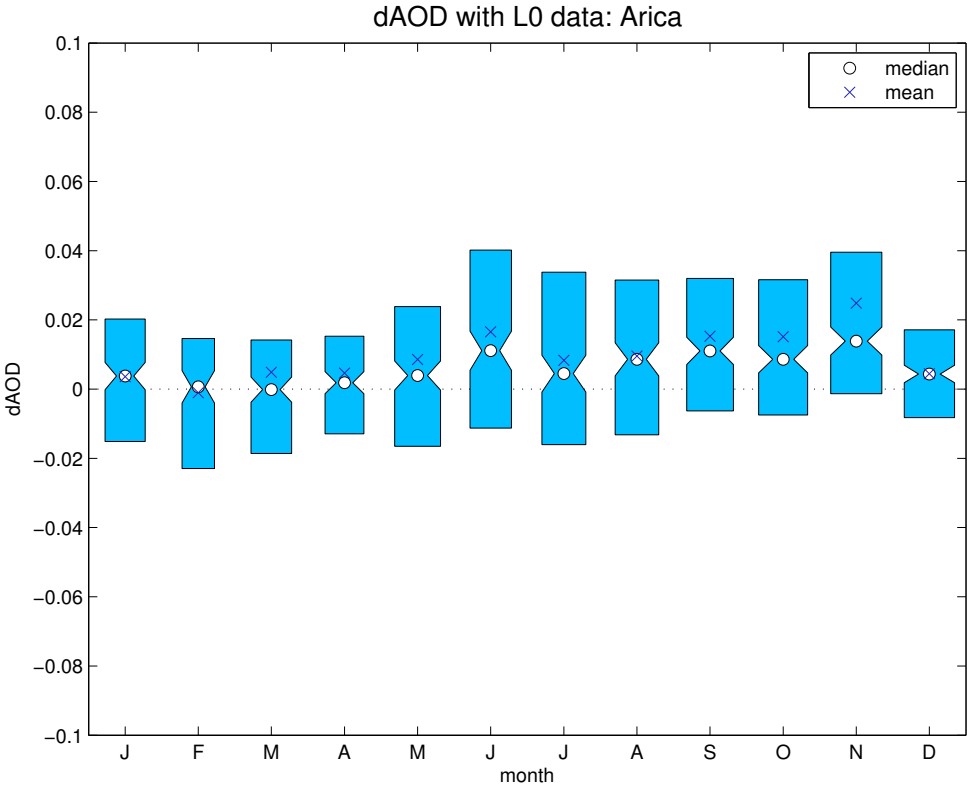

**Figure 6.** Monthly $dAOD$ (difference in L0 AOD and L2 AOD) for Arica, Chile.

As the cloud parcel model calculates the ambient/wet size distribution of aerosol and cloud droplets according to the aerosol dry size, in order to obtain the aerosol dry size distribution, we used the monthly mean AERONET-measured size distributions (August 2003) from the Level2 inversion product. The size distributions from AERONET represent the ambient size distributions so we initialized the model with a bi-modal dry size distribution that reproduced the observed ambient size distribution at a given relative humidity (RH).

For this purpose we needed to make several assumptions on the particle composition and vertical aerosol profiles. The first assumption to be made was to assume the chemical composition for the aerosol. For Lanai, we assumed the aerosol to be solely composed of NaCl to represent highly hygroscopic sea salt aerosol. For Walker Branch, the composition was assumed to be 50% insoluble organics and 50% inorganic ammonium sulfate. This composition is more representative of continental aerosol. The second assumption was that all aerosol was residing in the boundary layer. In




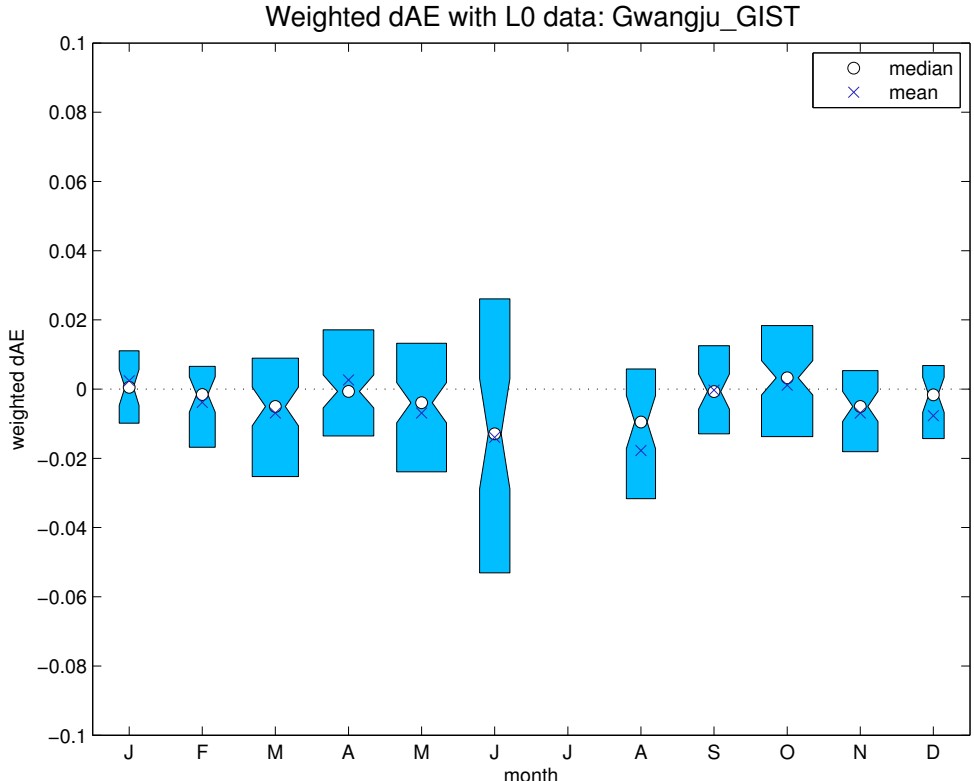

**Figure 7.** Similar to Figure 5, but for Gwangju Gist, Korea.

reality this might not hold, but does not affect the qualitative analysis performed here. The third

assumption was that the atmospheric RH profile and boundary layer height had to be estimated. This
is relevant especially in the case of hygroscopic aerosol: the higher RH we assume for cloud free
conditions, the smaller will be the corresponding particle dry size.

The cloud parcel model output is the ambient size distribution at each model level. These, in turn,
were used as an input to Mie calculations, in order to obtain extinction coefficient and corresponding

Angstrom Exponents at each model level and also the integrated columnar estimates. These calcu-
lations were carried out using the Mie calculation tool in LibRadtran (Mayer and Kylling, 2005),
assuming purely scattering particles with the real part of refractive index of 1.5.

Cloud parcel simulations were initialized using the dry size distribution determined for Lanai and
Walker Branch. Apart from the initial size and composition distribution, the initial conditions were

assumed to be the same in all simulations: ambient temperature of 288 K and RH of 63 %. After





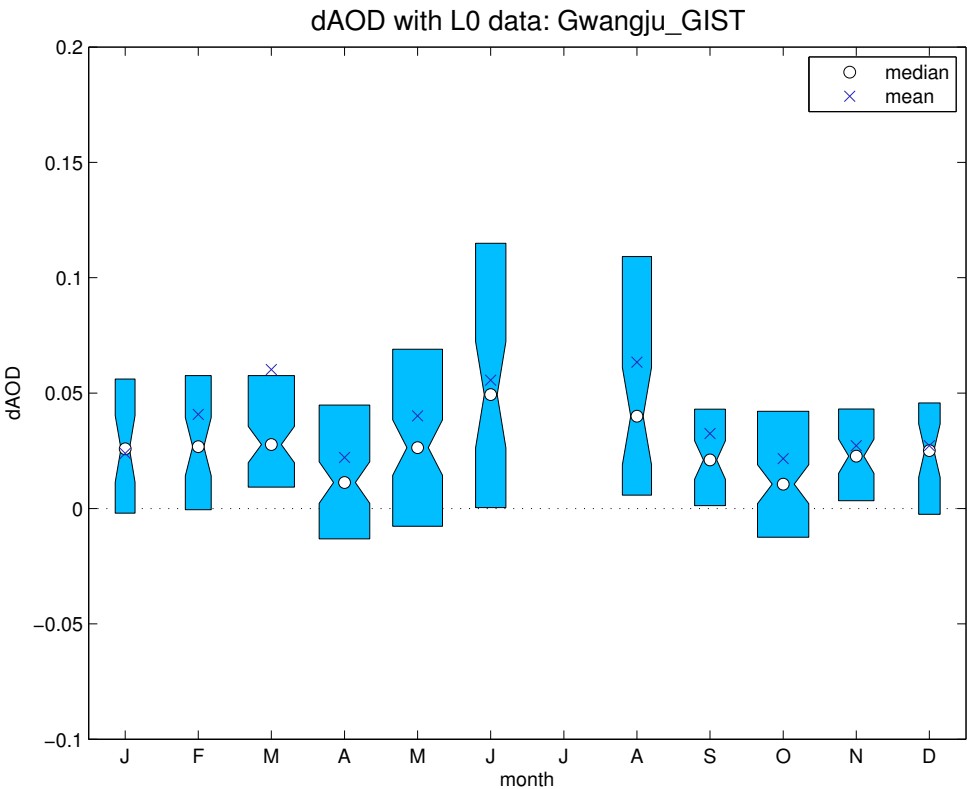

**Figure 8.** Similar to Figure 6, but for Gwangju Gist, Korea.

the initialization, model simulated cloud formation for an adiabatically ascending air parcel with a constant updraft velocity $w$. In Lanai, $w$ was assumed to be $0.2\,\mathrm{m\,s^{-1}}$ which is quite typical for marine stratocumulus clouds. In Walker Branch $w$ was assumed to be $0.5\,\mathrm{m\,s^{-1}}$. In the simulations the boundary height was assumed to be $1\,\mathrm{km}$. With the assumed adiabatic temperature profile the cloud depth was $100\,\mathrm{m}$ for Lanai.

From the simulated wet size distributions, we calculated the optical properties of the aerosol/cloud droplet population for different altitudes. To illustrate the effect of hygroscopicity on the aerosol optical properties, we used two different dry size distributions for Lanai: one, where the dry size distribution reproduced the AERONET-observed ambient size distribution at $80\,\%$ RH and one where it

does the same at $85\,\%$ RH. For the case where the size distribution was fitted at $80\,\%$, the growth factor (the ratio between effective mean size and dry size) is approximately 2 while at $85\,\%$ the growth factor is approximately 2.2 (see the Figure 1 of Ming and Russell (2001)). Thus, in the latter case,





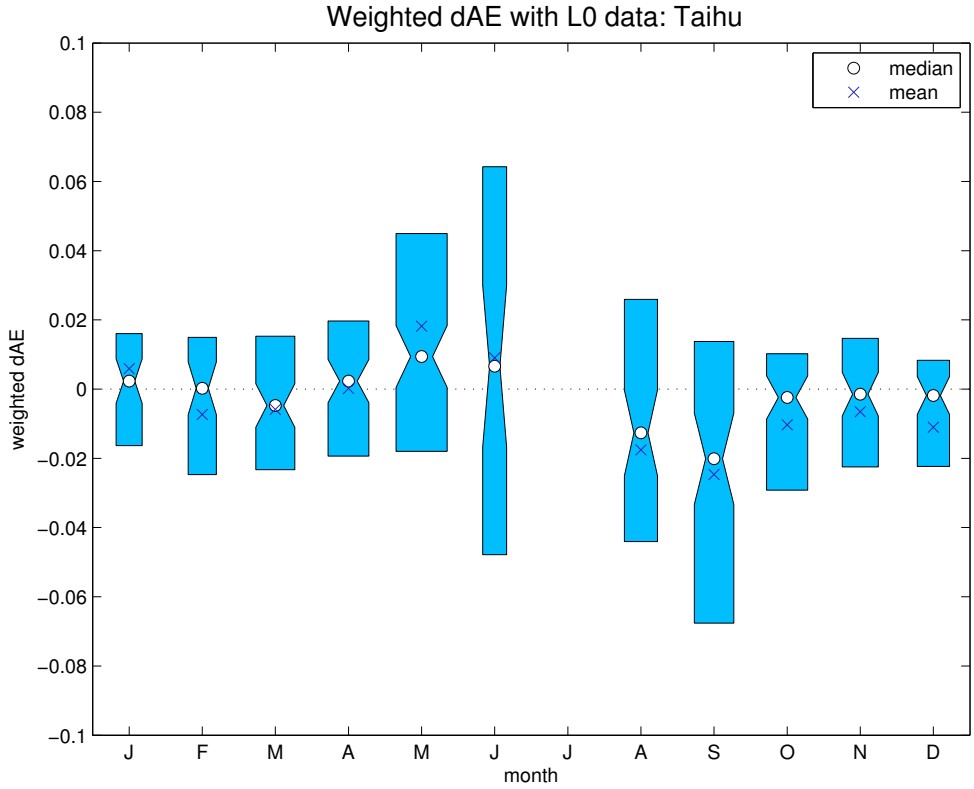

**Figure 9.** Similar to Figure 5, but for Taihu, China.

the dry size required to reproduce the AERONET observations was slightly smaller than in the case of 80 %. Figure 16 shows the calculated extinction and AE profiles for these two different cases.

The total column AOD and AE, as Cimel measurement would "see" from these profiles, were estimated both for the cloudy and clear-sky case. In the latter case, the column AOD and AE values were integrated over the model calculated aerosol profile from the ground level up to a level where RH reached 3 % higher value than RH that was used to determine the dry size distribution. This was thought to correspond to the highest humidity in clear-sky conditions. The remaining layers above

this level were integrated assuming that they have this constant extinction.

The numbers given within the Figure 16 give AOD and AE for clear-sky and cloudy sky conditions, as an estimate of the AERONET-measurement from these profiles. The uppermost three values correspond to the case of growth factor of 2 and lower three numbers to the case of growth factor of 2.2. This demonstrates that indeed the $dAE$ can turn to a positive value, for highly hygroscopic





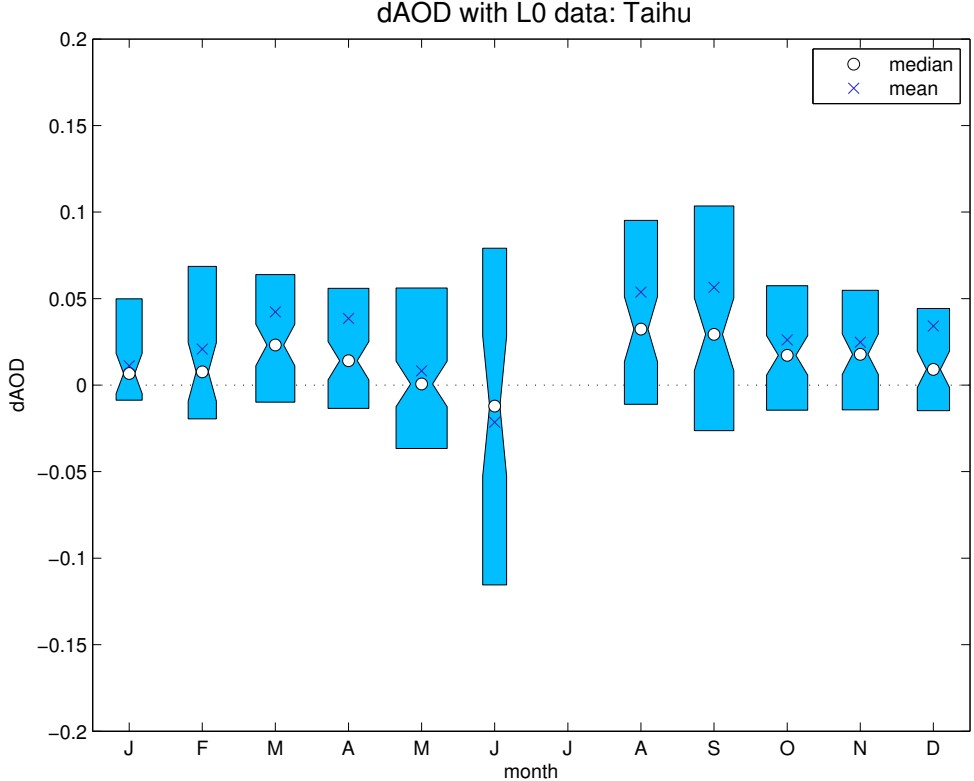

**Figure 10.** Similar to Figure 6, but for Taihu, China.

NaCl aerosols, and when the particle dry sizes are sufficiently small and the supersaturation at the cloud base is high enough to activate these particles. In such a case the largest interstitial aerosol particles are clearly smaller than the estimated size limit for fine mode aerosol classification. For less hygroscopic aerosol composition and higher total aerosol concentration, e.g. the one we assumed for our Walker Branch simulations, did not produce positive $dAE$ cases. In Lanai, the activation of smaller particles than in Walker Branch was further assisted by low number concentration and small dry sizes of aerosol which allowed the maximum supersaturation to reach higher values as were not very efficient in depleting gas phase water during cloud activation.

Based on our model simulations, it seems evident that cloud activation can affect and remove particles from the AERONET-measured fine mode AOD, resulting in positive $dAE$ between L0 and L2 measurements. However, this requires highly hygroscopic aerosol composition (sea salt) with sufficiently small dry sizes. We wanted to additionally assess this threshold of activation size





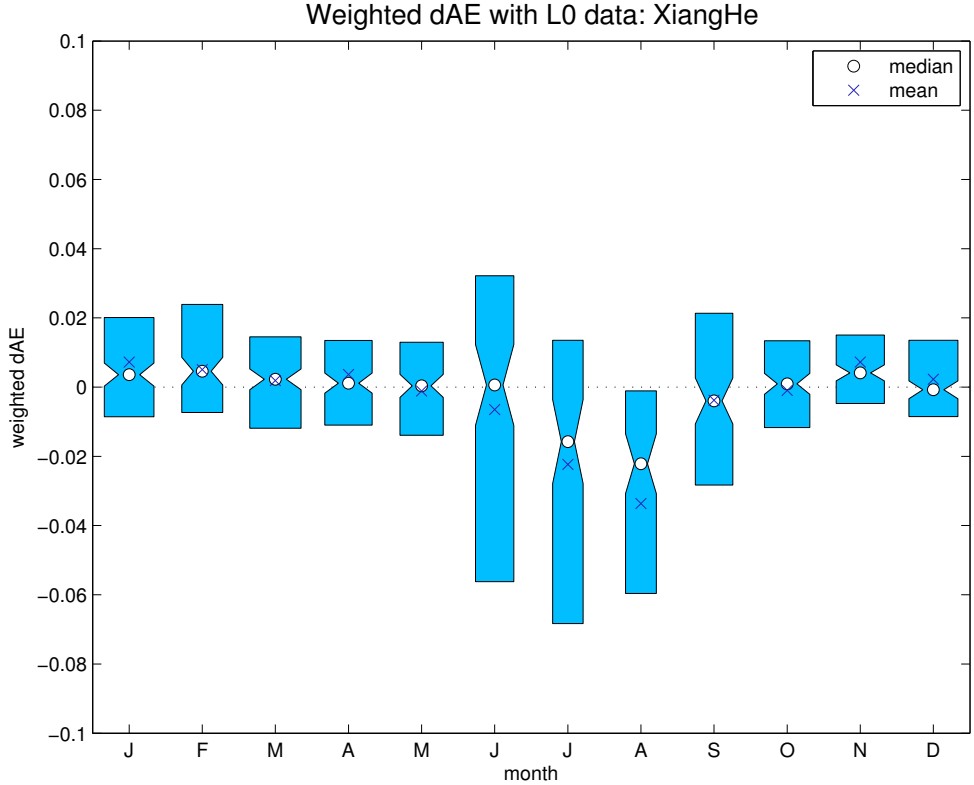

**Figure 11.** Similar to Figure 5, but for XiangHe, China.

that needs to be reached, so that effect by the removal of the cloud activated largest size particles in the fine mode, would overcome the hygroscopic growth of the smaller particles in the case of marine aerosol size distribution observed in Lanai. The former process has the overall effect to

increase AE in cloudy case, while the latter has an opposing effect. The upper plot of the Figure 17 shows the cloud activation (critical) radius as a function critical supersaturation. We repeated our Mie simulations for a range of critical supersaturation from 0 to 0.3%, always removing particles larger than the critical radius corresponding to the critical supersaturation from the aerosol size distribution. The lower panel in Fig 17 shows $dAE$ as a function critical supersaturation. The figure illustrates

that it is very evident that only a relatively narrow range of critical supersaturation, corresponding to critical radius of about 0.3 $\mu$m, results in positive AE difference between cloudy and clear-sky column. With high supersaturations such small particles activate that interstitial aerosol is no longer



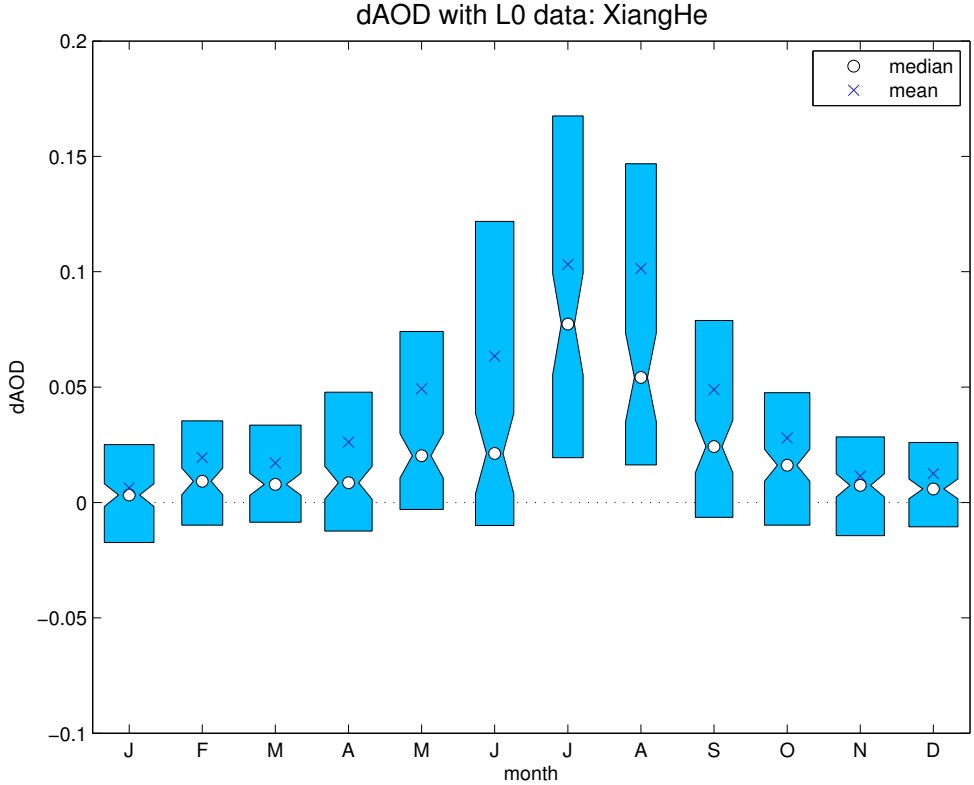

**Figure 12.** Similar to Figure 6, but for XiangHe, China.

affecting column AOD and thus AE. At low supersaturations, the hygroscopic growth of interstitial particles dominates the column AE, and thus the influence of a cloud results in negative $dAE$.

Figure 18, in turn, shows the results from our simulations in more detail for several cases. The simulations of both Walker Branch and Lanai are shown at different model levels, e.g. WB@820 refers to the Lanai simulation at the model altitude of 820m, thus at the base of the cloud. The results close to the cloud top at 970 m are also shown ("Lanai@970" and "WB@970" for Lanai and Walker Branch, respectively). Finally, for the case of Lanai and close to the cloud top, we included

two cases of different aerosol dry sizes, as discussed above and were shown in the Figure 16. The left-hand-side y-axis shows these cases and the total particle area.

    We also estimated AE for a range of single effective sizes by assuming a very narrow log-normal size distribution to represent a mono-disperse case for a size range up to 0.6 $\mu$m, this is shown by a black line corresponding to the right-hand-side y-axis. This AE estimate was calculated from our





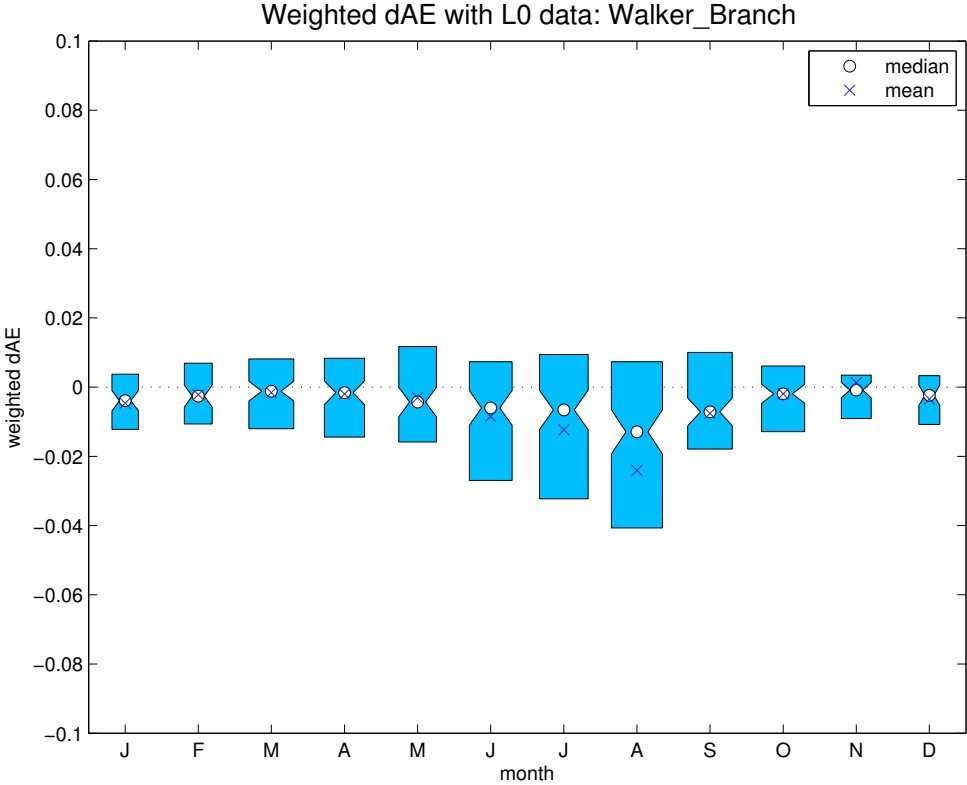

**Figure 13.** Similar to Figure 5, but for Walker Branch, TN, USA.

modeled extinction efficiencies at 500nm and 495nm. Since the extinction efficiency multiplied by
the total particle area gives the total extinction, this choice of plots in RHS and LHS y-axes gives an
opportunity to assess the impact of different particle sizes in the overall AE. The total area plots can
be considered as weights from different sizes in the overall AE.

Close to the cloud bottom (at 820 m) there is no cloud activation, while the main interest is to see
the patterns at higher in the cloud at 970 m: the activation size limit in the Walker Branch simulation
is at around 0.4 $\mu$m and lower in Lanai cases. As demonstrated in the Figure 16, the growth factor of
2.2 was required for our NaCl simulation to produce positive $dAE$ and indeed with the help of the
Figure 16; the sizes below about 0.3 $\mu$m (also the upward branch of extinction efficiency vs. particle
size for visible wavelength typically shown in in a textbook examples of Mie theory results) are
affecting to the positive AE. For instance, for Walker Branch there is substantial amount of weight
for sizes above this limit, even close to the cloud top. On the other hand, if the activation size reaches





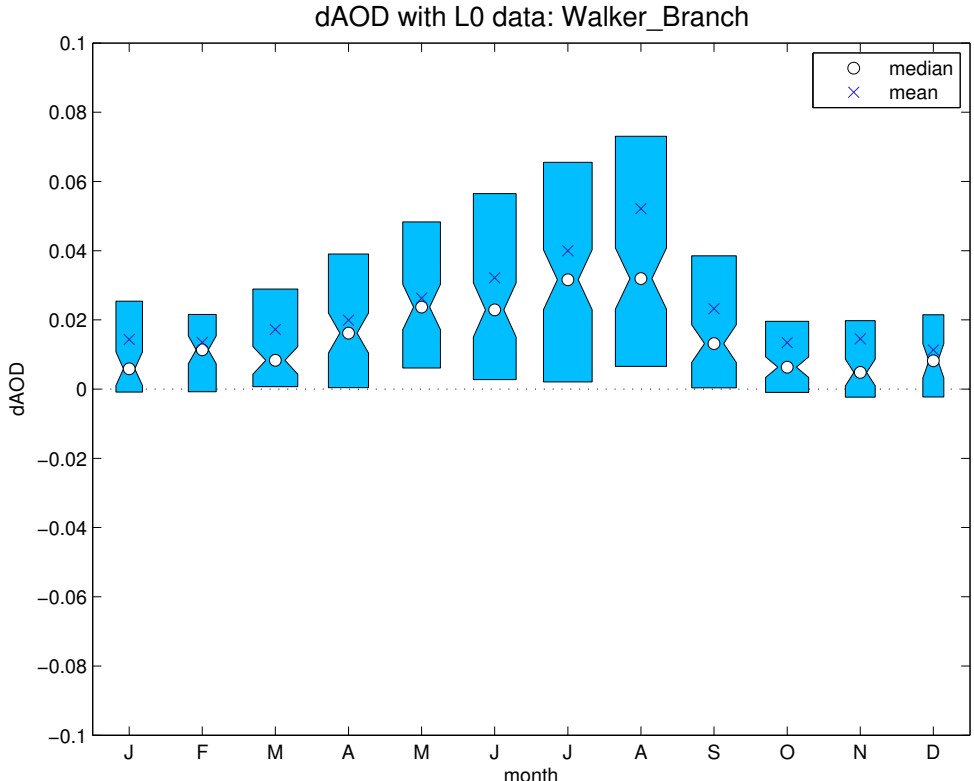

**Figure 14.** Similar to Figure 6, but for Walker Branch, TN, USA.

values close to this critical size, as is the case for Lanai with the growth factor of 2.2, then indeed
the chance to get positive AE becomes evident.

## 4   Conclusions

The studies of aerosol–cloud interactions, exploiting remote sensing measurements, are challenging
and therefore many aspects have remained poorly known. Typically the aerosol optical properties
can be measured by passive remote sensing approaches only for clear-sky conditions. Active remote
sensing (mainly Lidar) does not suffer equally about cloud adjacency effects, however the coverage
that one can reach currently by active remote sensing is more limited. There exists one remote

sensing product, so called spectral deconvolution (SDA) from AERONET, that can offer a unique
information about the cloud effect on AOD. Therefore, it is somewhat surprising that these data have





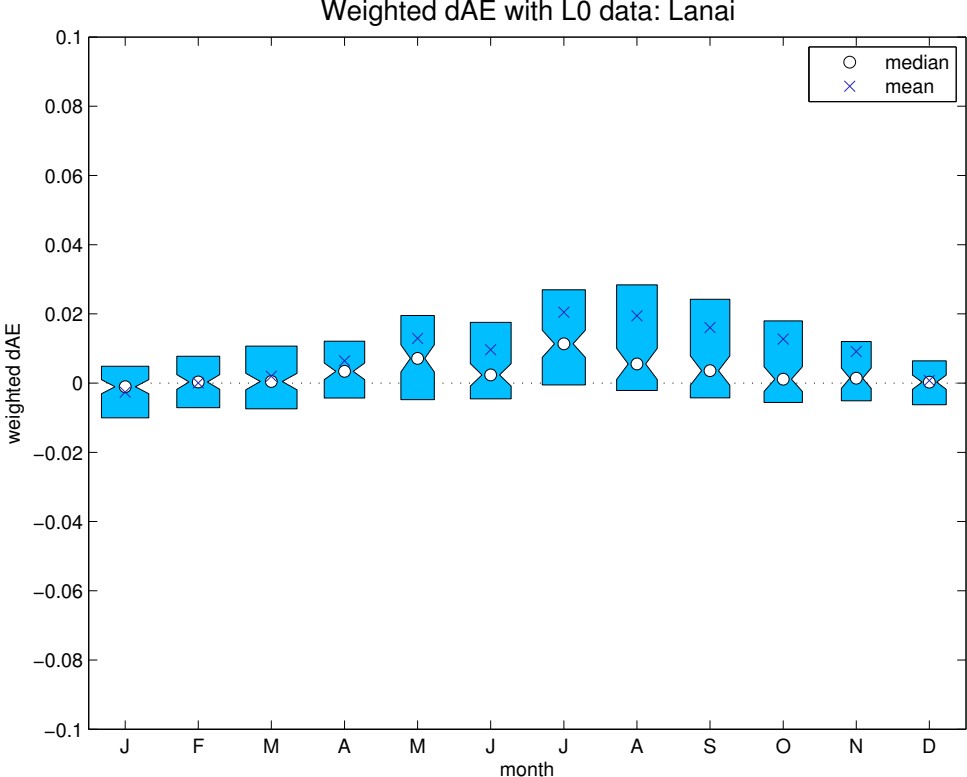

**Figure 15.** Similar to Figure 5, but for Lanai, Hawaii, USA.

not been yet fully exploited for this purpose. We analyzed SDA for different cloud conditions to give quantitative estimates for the cloud enhanced AOD values, using all the available AERONET sites.

We performed the analysis on a seasonal basis and found that consistently the highest cloud related 300 AOD enhancements occur in sites in East-Asia, reaching levels of AOD of about 0.1. In relative terms, these values are in range of 10-12% higher if compared to clear-sky (Level2) fine mode AOD. This is not insignificant and should be taken into account for, e.g. in the calculations of aerosol radiative effects. On the other hand, the difference over all the included sites is rather notable as well, e.g. in JJA fine mode AOD of all-sky data is 0.011 higher than the mean based on Level2 only 305 (0.154), thus all-sky fine mode AOD being about 7% higher.

We estimated similarly the differences in fine mode AE, between cloudy- and clear-sky cases. In majority of the cases, negative AE differences were typically prevailing. These cases are likely dominated by particle growth in the humid conditions over the cloud activation. There were only





about ten sites of clearly positive $dAE$, all being strongly affected by marine aerosols. It is noted
that the AE changes were rather small, only few percent. Small new accumulation particles from both
growth of Aitken sized particles and gas-to-particle conversion may counterbalance humidification
growth of some existing accumulation mode particles, thereby resulting in little change in AE.

Albeit overall $dAE$ was small, in the marine cases the negative $dAE$ cases were essentially missing, thus suggesting that different processes dominate if compared to the continental cases. Therefore, we investigated in more detail, with the help of cloud parcel model, the relative role of aerosol
hygroscopic growth and cloud activation in different cloudy conditions. Our model simulations supported that cloud activation can affect and remove particles from the AERONET-measured fine mode
AOD, resulting in positive $dAE$ between L0 and L1 measurements. However, this requires highly
hygroscopic aerosol composition (sea salt) with sufficiently small dry sizes.

*Acknowledgements.*



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



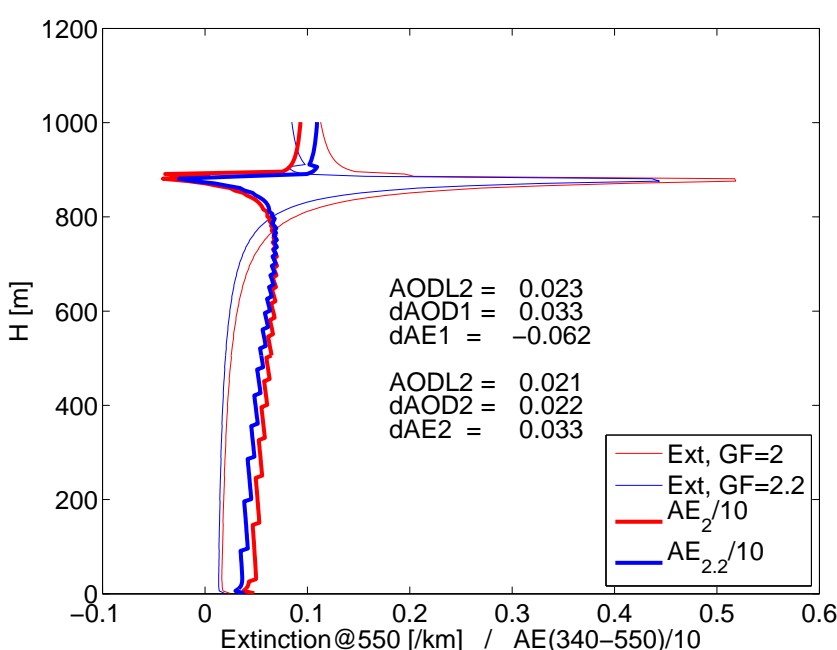

**Figure 16.** Profile of extinction at 500nm and Angstrom Exponent from 340-500nm wavelength pair. The latter is divided by 10, in order to better match the x-scale.



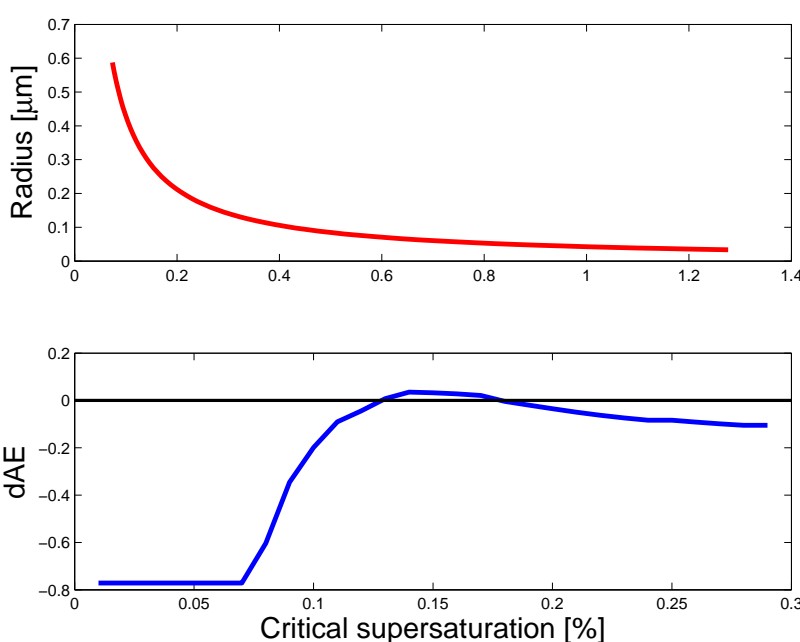

**Figure 17.** Critical cloud droplet activation radius as a function of critical supersaturation (upper plot) and estimated difference in AE, between cloudy- and clear-sky case (lower plot).





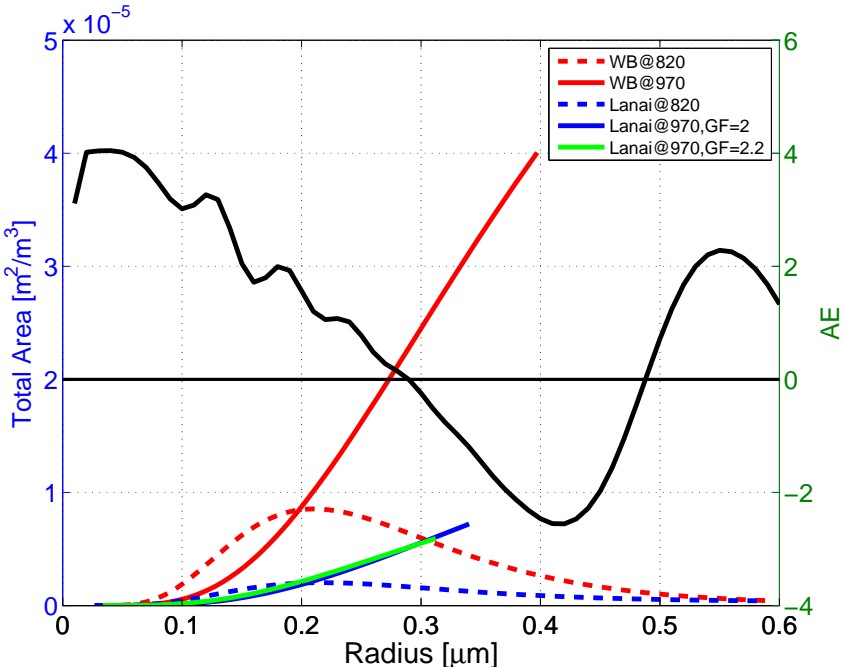

**Figure 18.** Total aerosol area per volume (RHS y-axis) at different model levels, WB refers to Walker Branch, e.g. WB@820 means Walker Branch case and altitude of 820m. Lanai is shown for two growth factors, 2 and 2.2, as discussed in more details in the text. LHS y-axis shows Angstrom Exponent, as a estimated for mono-disperse aerosol of given radius.



**Table 1.** Seasonal AOD based on sampling for different cloudiness. L1 and L2 refer to Level1 and Level2 of AERONET data, respectively. L0 refers to those cases of L1, which did not belong to L2, thus cloudy cases only. Different regions have the following abbreviations: NAm (North America), SAm (South America), Eu (Europe), NAf (North Africa), SAf (South Africa), As (Asia), Aus (Australasia). These regions are indicated by the solid lines in the Figures 1-4.

| NAm | | | | | SAm | | | |
|---|---|---|---|---|---|---|---|---|
| AOD | DJF | MAM | JJA | SON | DJF | MAM | JJA | SON |
| L1 | 0.052 | 0.096 | 0.147 | 0.078 | 0.088 | 0.065 | 0.126 | 0.249 |
| L2 | 0.047 | 0.088 | 0.137 | 0.073 | 0.074 | 0.057 | 0.119 | 0.237 |
| L0 | 0.059 | 0.105 | 0.158 | 0.087 | 0.095 | 0.071 | 0.139 | 0.262 |
| d12 | +0.005 | +0.008 | +0.010 | +0.005 | +0.014 | +0.008 | +0.007 | +0.012 |
| d02 | +0.012 | +0.017 | +0.021 | +0.014 | +0.020 | +0.014 | +0.020 | +0.025 |

| Eu | | | | | NAf | | | |
|---|---|---|---|---|---|---|---|---|
| AOD | DJF | MAM | JJA | SON | DJF | MAM | JJA | SON |
| L1 | 0.092 | 0.137 | 0.152 | 0.108 | 0.124 | 0.112 | 0.146 | 0.138 |
| L2 | 0.089 | 0.129 | 0.144 | 0.103 | 0.120 | 0.108 | 0.142 | 0.134 |
| L0 | 0.101 | 0.149 | 0.168 | 0.118 | 0.136 | 0.122 | 0.159 | 0.153 |
| d12 | +0.003 | +0.008 | +0.009 | +0.005 | +0.005 | +0.004 | +0.004 | +0.004 |
| d02 | +0.013 | +0.020 | +0.025 | +0.015 | +0.016 | +0.014 | +0.017 | +0.019 |

| SAf | | | | | As | | | |
|---|---|---|---|---|---|---|---|---|
| AOD | DJF | MAM | JJA | SON | DJF | MAM | JJA | SON |
| L1 | 0.126 | 0.084 | 0.141 | 0.168 | 0.289 | 0.319 | 0.262 | 0.268 |
| L2 | 0.101 | 0.074 | 0.135 | 0.160 | 0.278 | 0.304 | 0.240 | 0.254 |
| L0 | 0.139 | 0.092 | 0.150 | 0.174 | 0.303 | 0.332 | 0.273 | 0.282 |
| d12 | +0.026 | +0.010 | +0.006 | +0.008 | +0.011 | +0.015 | +0.022 | +0.014 |
| d02 | +0.038 | +0.018 | +0.015 | +0.014 | +0.025 | +0.029 | +0.033 | +0.028 |

| Aus | | | | |
|---|---|---|---|---|
| AOD | DJF | MAM | JJA | SON |
| L1 | 0.095 | 0.093 | 0.110 | 0.140 |
| L2 | 0.085 | 0.083 | 0.099 | 0.129 |
| L0 | 0.102 | 0.098 | 0.118 | 0.148 |
| d12 | +0.010 | +0.010 | +0.011 | +0.011 |
| d02 | +0.016 | +0.016 | +0.019 | +0.019 |



**Table 2.** Otherwise similar to Table 1, but showing overall results for all the sites.

| All sites | | | | |
|---|---|---|---|---|
| AOD | DJF | MAM | JJA | SON |
| L1 | 0.119 | 0.145 | 0.165 | 0.143 |
| L2 | 0.112 | 0.136 | 0.154 | 0.135 |
| L0 | 0.128 | 0.155 | 0.178 | 0.154 |
| d12 | +0.007 | +0.009 | +0.011 | +0.008 |
| d02 | +0.016 | +0.020 | +0.024 | +0.018 |