# Peer review of "Assessment of cloud related fine mode AOD enhancements based on AERONET SDA product"

_Atmospheric Chemistry and Physics, 2016_

## Short Comment (SC1)

We thank the reviewer for his/her constructive comments. We will provide a comprehensive revision and point-by point response later, while the purpose of this short note is just to clarify one apparent misconception that has now appeared. Since it is about a very essential and fundamental point, regarding the basis of our study, we considered it useful to clarify this misconception already now, at this stage of the revision processes. It is about the longish point #1 of the reviewer, however its main message can be perhaps best summarized by this sentence: "we don't know how good those retrievals are and how much they are contaminated by cloud". In other words, the reviewer had two major concerns: 1) the quality of the L1 measurements we included, 2) the ability of the SDA fine mode AOD to represent aerosol optical depth in cloudy conditions. We try to explain both issues below.

**First about the data quality of L1 and L2 for the purpose of our study.** There was indeed a very careful checking of the retrievals included, as we will further elaborate below. Many of these things take place already in the normal AERONET data processing. Moreover, we applied several additional criteria regarding what Level1 data to include. We admit that the latter points, in particular, were not sufficiently stressed in the current version of the manuscript and we will improve the revised manuscript in this respect.

First, briefly about the "AERONET-inboard" checking. Pertinent here is, for example, an excerpt from Eck et al. 2014: "The direct sun measurement data are not included in the AERONET Level 1.0 data set if the variance of the raw signal is very high within the triplet sequence. The variance threshold applied is based on the root mean square (RMS) differences of the three direct sun triplet measurements relative to the mean of these three values. If the (RMS/mean)·100% of the triplet values is greater than 16% then the data will not be used for computation of AOD and the data will not appear in the Level 1.0 data set. This temporal variance threshold primarily removes data that are affected by clouds with large spatial–temporal variance in COD. This effectively removes much of the cumulus cloud contaminated data, although some of the thinner edges with lower COD do remain in the data."

In the AERONET Version 2 Level 2 database, if there are only 1 or 2 points remaining in a day after automatic cloud screening (Level 1.5), then none of this data reaches Level 2. In other words, at least 3 AOD observations are need to pass the Smirnov et al. (2000) cloud screening algorithm in order for the data to reach Level 2 for that day.

Second, briefly about the criteria (that we will thoroughly describe and include also in the revised version) that we applied in our additional QA checking: 1) we required that AOD for SDA algorithm was available from all the channels utilized by SDA (380, 440, 500, 675, and 870 nm), in order to ensure that the AOD spectra input to the SDA were always of good quality, 2) Level 2 data had to be available within one-week time window, to rule out any instrumental problems, 3) outliers were removed according to the following criterion:

$\text{Abs}(AOD500nm - AODSDA500nm) > (0.02 + AOD500nm*0.005)$.

NOTE: This is the same consistency check between measured AOD at 500 nm and SDA retrieved total AOD at 500 nm that is applied in the quality control checks for AERONET Level 1.5 data for SDA.

We believe that after these QA steps, the retrievals that were eventually selected were indeed thoroughly checked to include meaningful information for the purpose of our study.

**Second, briefly about the ability of SDA fine mode AOD to represent AOD in cloudy conditions.**
Eck et al. 2014, that we cited and referred to as well, includes a lot of information/justification why the AERONET Version 2 Level 1 data include meaningful information for this type of analysis that we have carried out. Figure 3 there, in conjunction with Figures 16a and 16b, show how the large triplet variation data, which is often screened from Level 2, is in fact good fine mode AOD data, but with higher temporal variability due to the turbulent and dynamic conditions in the vicinity of cumulus clouds.

The following direct excerpt from Eck et al. 2014 will hopefully further clarify this issue: "The AERONET data in Fig. 2 were not screened for clouds (Level 1; see Sect. 2.2 below), since O'Neill et al. (2003) have shown that SDA identifies cloud optical depth as the coarse mode AOD component. Analysis by Chew et al. (2011) of AERONET measured spectral AOD in conjunction with lidar data in Singapore has shown that the SDA technique effectively separated the coarse mode (cirrus cloud contamination, as identified by lidar) from the total optical depth without affecting the fine mode component. Additionally, Kaku et al. (2014) have verified that the SDA technique is also effective in separating the fine and coarse modes from in situ spectral optical measurements."

REFERENCES

Eck, T. F., Holben, B. N., Reid, J. S., Arola, A., Ferrare, R. A., Hostetler, C. A., Crumeyrolle, S. N., Berkoff, T. A., Welton, E. J., Lolli, S., Lyapustin, A., Wang, Y., Schafer, J. S., Giles, D. M., Anderson, B. E., Thornhill, K. L., Minnis, P., Pickering, K. E., Loughner, C. P., Smirnov, A., and Sinyuk, A.: Observations of rapid aerosol optical depth enhancements in the vicinity of polluted cumulus clouds, Atmos. Chem. Phys., 14, 11633-11656, doi:10.5194/acp-14-11633-2014, 2014.

---

## Referee Comment (RC1) · Anonymous Referee #1 · 23 Aug 2016

This manuscript uses AERONET AOD products retrieved from the spectral deconvolution algorithm to 1) quantify the AOD enhancement in cloudy-sky conditions, 2) examine the change in Angstrom exponent due to cloudy-sky, and then 3) propose potential explanations that are responsible for the change in Angstrom exponent. While I find the scope of the work is interesting, I have some concerns.

1) The use of "Level 0", the data points that got removed from Level 1 and was not included in Level 2 due to cloud screening:

The whole manuscript is based on the assumption (although the authors treated it as a fact) that the fine- mode retrieval that is in Level 1 but not in Level 2 represents aerosol properties in cloudy conditions. The problem is - those retrievals were removed at the first place because we don't know how good those retrievals are and how much they are

contaminated by clouds. The authors argue that clouds affect coarse-mode retrievals but not fine-mode, and therefore, the use of fine-mode retrieval is OK. However, I don't see any evidence to show that those retrievals are valid and indeed representative. The paper they cited (Chew et al.) used level 1.5, not level 1, so Chew's conclusions shouldn't be applied directly without caution. In short, as the authors mentioned, it is somewhat surprising that these data have not been fully exploited, but there is a reason for that. It is not scientifically rigorous to use retrievals without checking if those retrievals are meaningful!

Minor comment – Page 4: do the authors really mean "the latter for all-sky and the former for clear-sky"? I think it should the other way around.

2) The analysis of seasonal variation and significance:

It would be better to clearly describe the sample size used in each bin, and to include retrieval uncertainty into these analyses. While the data range (figures 5-15) in each month spreads quite widely, it would improve the manuscript greatly by providing more critical discussions about them, rather than simply focusing on means only. Also, the authors throw in something like "For cumulus clouds in the mid-Atlantic US, the AE. . ." or "marine stratocumulus" for Lanai, which needs more care; these statements should be supported by some scientific evidence (a quick way will be to check weather state from ISCCP).

3) Parcel model runs

I thought this part is interesting, but the current descriptions lack logical connections and are very dis-organised. I don't think readers can replicate simulations/results based on the current form, and I would strongly recommend rewriting this part. Here are some specific examples, which hopefully can help the authors understand why the current form could be quite confusing and unclear.

a) Could the authors make it clearer about the initial size and composition distribution

used in the simulations? Like, the sentence on Page 14, 'For less hygroscopic aerosol composition. . .., e.g., the one we assumed for our Walker Branch simulations", which should be described clearly right in the beginning. What is the growth factor used for Walker Branch? Also, on Page 16, it is mentioned that a very narrow lognormal size distribution, but I don't recall what is used exactly in the control experiments? Perhaps it is mentioned somewhere in the manuscript, but these things should be introduced in a more coherent and organised manner.

b) Could the authors make it clearer how the total column AOD and AE are calculated in cloudy and clear-sky conditions? A lot of assumptions are made there and it is unclear where is the justification, and why this will be consistent to observations.

c) Page 17: why using different combinations of wavelengths for Angstrom exponent calculations?

d) The context of "For instance, for Walker Branch there is . . .for sizes above this limit, even close to the cloud top" on Page 17? Also, which part of Figure 18 helps conclude the last sentence of section 3?

e) Errors on Page 16, "WB@820 refers to the Lanai simulation", legend in Figure 16, and captions for Figure 18.

---

## Referee Comment (RC2) · Anonymous Referee #2 · 6 Oct 2016

This paper describes an interesting new approach to assess the AOT for fine mode aerosols in cloudy observations from AERONET. Instead of screening for clouds and retrieving fine and coarse mode AOT, the coarse mode AOT in aerosol-cloud mixed observations is attributed to cloud optical thickness, giving the fine mode AOT in cloud conditions. Given this the study tries to assess the magnitude of cloud enhancement in the fine mode.

While the approach is interesting, the paper lacks a sound scientific approach and clear presentation. In my opinion there is a need for a better description of the new approach and the expected impact, results to show the new approach works as expected, or not, and a conclusion. Instead many measurements are presented, explanations are presented, but it is not clear whether the method by itself can be trusted or not. Instead a large number of measurements are shown, which by itself do not prove the underlying

new approach.

The paper should be restructured so to follows a clear scientific approach: state the problem, describe how this problem is going to be resolved and then discussion and conclusion. In it's current form there are too many figures, which do not add to the understanding of the problem, and new discussions are started in the middle of the manuscript. Furthermore, the text is sometimes hard to follow due to strange reasoning or formulations.

Specific comments are below:

line 50: "..(e.g. due to the aqueous process including nitrate or sulfate".

This is a strange sentence: What processes are meant here? "Nitrate" and "Sulphate" are not 'processes'.

lines 59-61: Based on its assumptions, SDA identifies cloud optical depth as the coarse mode AOD component and therefore effectively computes the fine mode AOD also in mixed cloud-aerosol observations.

The assumptions used by the SDA algorithm are not discussed, however, they are vital to assess how this new method works. E.g., from this sentence it is not even clear whether SDA attributes coarse mode AOT to clouds in only mixed phase conditions or all conditions. In the latter case, does the fine mode AOT relate well to the original fine mode AOT (in cloud-free observations)?

line 65: AERONET SDA product has been used to some extent, i.e. for rapid AOD increases in the vicinity of cumulus.

Again an unclear sentence. Do you mean to say that SDA has been used 'to study' rapid AOD increases?

lines 82-86: The spectral deconvolution algorithm (SDA) product, and its ability to separate coarse and fine mode AOD and provide useful fine mode AOD also in cloudy

conditions, is vitally important in our study. O'Neill et al. (2001, 2002) developed SDA algorithm that utilizes spectral total extinction AOD data, ...

These sentences are bad English. Please rephrase.

line 98: insert 'mode' after 'coarse'

line 99: remove the comma

lines 106-107: we included fine mode AOD and AE at 500 nm, from both Level1 and Level2 SDA measurements, the latter for all-sky conditions and the former for clear-sky conditions.

I'm not sure about this one, but I expect this to be the other way around: L1 being all-sky and L2 only clear-sky.

lines 122-130 should be part of the Introduction, not results.

line 145 and figures 5 to 14. There are far too many figures here that do not add too much information. These figures should be merged into two Figures maximum.

Line 299: We performed the analysis on a seasonal basis and found that consistently the highest cloud related 300 AOD enhancements occur in sites in East-Asia..

Please, write concisely, these kind of statements make the manuscript very hard to read: make an analysis on the basis of seasonality and find a region that stands out.

---

## Author Comment (AC1) · 7 Dec 2016

We would like to first express our thanks to the **REFEREE #1** for his/her constructive comments. The responses to these are below after the reviewer points that are in bold.

**This manuscript uses AERONET AOD products retrieved from the spectral deconvolution algorithm to 1) quantify the AOD enhancement in cloudy-sky conditions, 2) examine the change in Angstrom exponent due to cloudy-sky, and then 3) propose potential explanations that are responsible for the change in Angstrom exponent. While I find the scope of the work is interesting, I have some concerns.**

**1) The use of "Level 0", the data points that got removed from Level 1 and was not included in Level 2 due to cloud screening: The whole manuscript is based on the assumption (although the authors treated it as a fact) that the fine- mode retrieval that is in Level 1 but not in Level 2 represents aerosol properties in cloudy conditions. The problem is - those retrievals were removed at the first place because we don't know how good those retrievals are and how much they are contaminated by clouds. The authors argue that clouds affect coarse-mode retrievals but not fine-mode, and therefore, the use of fine-mode retrieval is OK. However, I don't see any evidence to show that those retrievals are valid and indeed representative. The paper they cited (Chew et al.) used level 1.5, not level 1, so Chew's conclusions shouldn't be applied directly without caution. In short, as the authors mentioned, it is somewhat surprising that these data have not been fully exploited, but there is a reason for that. It is not scientifically rigorous to use retrievals without checking if those retrievals are meaningful!**

We provided a brief note already earlier during the open discussion regarding this reviewer point. Many of these clarifications/justifications are now included also in our revised manuscript to explain the quality assurance included in the AERONET measurement data set that we analyzed and thus to justify their use in our study. We also included some illustrative plots, both in the manuscript and in the Supplement, to indicate how the L1 fine mode AOD is indeed a meaningful measurement also in cloudy conditions.

**Minor comment – Page 4: do the authors really mean "the latter for all-sky and the former for clear-sky"? I think it should the other way around.**

This is right, it should have been the other way around. This has been now corrected in the revised manuscript.

**2) The analysis of seasonal variation and significance:**
**It would be better to clearly describe the sample size used in each bin, and to include retrieval uncertainty into these analyses. While the data range (figures 5-15) in each month spreads quite widely, it would improve the manuscript greatly by providing more critical discussions about them, rather than simply focusing on means only. Also, the authors throw in something like "For cumulus clouds in the mid-Atlantic US, the AE ... " or "marine stratocumulus" for Lanai, which needs more care; these statements should be supported by some scientific evidence (a quick way will be to check weather state from ISCCP).**

We included the sample size in each bin. Also, the discussion of the monthly plots is now more thorough.

**3) Parcel model runs**
**I thought this part is interesting, but the current descriptions lack logical connections and are very dis-organised. I don't think readers can replicate simulations/results based on the current form, and I would strongly recommend rewriting this part. Here are some specific examples, which hopefully can help the authors understand why the current form could be quite confusing and unclear.**
**a) Could the authors make it clearer about the initial size and composition distribution used in the simulations? Like, the sentence on Page 14, 'For less hygroscopic aerosol composition …, e.g., the one we assumed for our Walker Branch simulations", which should be described clearly right in the beginning. What is the growth factor used for Walker Branch? Also, on Page 16, it is mentioned that a very narrow lognormal size distribution, but I don't recall what is used exactly in the control experiments? Perhaps it is mentioned somewhere in the manuscript, but these things should be introduced in a more coherent and organised manner.**

The modeling section was strongly modified and re-structured. Therefore these points above, as well as the modeling related reviewer points below, are hopefully adequately addressed by our new revised manuscript.

**b) Could the authors make it clearer how the total column AOD and AE are calculated in cloudy and clear-sky conditions? A lot of assumptions are made there and it is unclear where is the justification, and why this will be consistent to observations.**

The model calculations are described in more detail in the revised manuscript.

**c) Page 17: why using different combinations of wavelengths for Angstrom exponent calculations?**

Good point. We agree that it is better to be consistent and changed the wavelengths in these runs. However, as expected, the pattern of AE as a function of size in this plot (and thus the conclusions) did not change.

**d) The context of "For instance, for Walker Branch there is … for sizes above this limit, even close to the cloud top" on Page 17? Also, which part of Figure 18 helps conclude the last sentence of section 3?**

These points are clarified in the revised manuscript.

**e) Errors on Page 16, "WB@820 refers to the Lanai simulation", legend in Figure 16, and captions for Figure 18.**

These are corrected.

---

## Author Comment (AC2) · 7 Dec 2016

We would like to first express our thanks to the **REFEREE #2** for his/her constructive comments. The responses to these are below after the reviewer points that are in bold.

**This paper describes an interesting new approach to assess the AOT for fine mode aerosols in cloudy observations from AERONET. Instead of screening for clouds and retrieving fine and coarse mode AOT, the coarse mode AOT in aerosol-cloud mixed observations is attributed to cloud optical thickness, giving the fine mode AOT in cloud conditions. Given this the study tries to assess the magnitude of cloud enhancement in the fine mode.**

**While the approach is interesting, the paper lacks a sound scientific approach and clear presentation. In my opinion there is a need for a better description of the new approach and the expected impact, results to show the new approach works as expected, or not, and a conclusion. Instead many measurements are presented, explanations are presented, but it is not clear whether the method by itself can be trusted or not. Instead a large number of measurements are shown, which by itself do not prove the underlying new approach.**

We provided a brief note already earlier during the open discussion regarding this point "whether the method by itself can be trusted or not". Many of these clarifications/justifications are now included also in our revised manuscript to explain the quality assurance included in the AERONET measurement data set that we applied and thus to justify their use in our study. We also included some illustrative plots, both in the manuscript and in the Supplement, to indicate how the L1 fine mode AOD is indeed a meaningful measurement also in cloudy conditions.

**The paper should be restructured so to follows a clear scientific approach: state the problem, describe how this problem is going to be resolved and then discussion and conclusion. In it's current form there are too many figures, which do not add to the understanding of the problem, and new discussions are started in the middle of the manuscript. Furthermore, the text is sometimes hard to follow due to strange reasoning or formulations.**

We have improved the manuscript as the reviewer suggested. There are only limited number of plots (of key results) in the actual manuscript, while many of the earlier plots are now in the Supplement. The text itself has been also clarified.

**Specific comments are below:**

**line 50: "..(e.g. due to the aqueous process including nitrate or sulfate". This is a strange sentence: What processes are meant here? "Nitrate" and "Sulphate" are not 'processes'.**

This is clarified. With these we mean the cloud processing occurring such as formation of sulphate in cloud droplets or redistribution of nitrate aerosol between different sized aerosol particles during cloud droplet formation and evaporation.

**lines 59-61: Based on its assumptions, SDA identifies cloud optical depth as the coarse mode AOD component and therefore effectively computes the fine mode AOD also in mixed cloud-aerosol observations.**

**The assumptions used by the SDA algorithm are not discussed, however, they are vital to assess how this new method works. E.g., from this sentence it is not even clear whether SDA attributes coarse mode AOT to clouds in only mixed phase conditions or all conditions. In the latter case, does the fine mode AOT relate well to the original fine mode AOT (in cloud-free observations)?**

In the revised version we have explained, and demonstrated by some example cases, how SDA works in difference cases of cloudiness. These examples cover cirrus clouds, when fine mode AOD is not cloud contaminated (case GSFC August 11, 2010; in the Supplement), as well as a example day of rapid cumulus variability (case BLDND from Dragon campaign, July 5, 2011; in the revised manuscript).

**line 65: AERONET SDA product has been used to some extent, i.e. for rapid AOD increases in the vicinity of cumulus. Again an unclear sentence. Do you mean to say that SDA has been used 'to study' rapid AOD increases?**

This sentence has been clarified.

**lines 82-86: The spectral deconvolution algorithm (SDA) product, and its ability to separate coarse and fine mode AOD and provide useful fine mode AOD also in cloudy conditions, is vitally important in our study. O'Neill et al. (2001, 2002) developed SDA algorithm that utilizes spectral total extinction AOD data, … These sentences are bad English. Please rephrase.**

These sentences have been rephrased.

**line 98: insert 'mode' after 'coarse'**

Done.

**line 99: remove the comma**

Done.

**lines 106-107: we included fine mode AOD and AE at 500 nm, from both Level1 and Level2 SDA measurements, the latter for all-sky conditions and the former for clear-sky conditions. I'm not sure about this one, but I expect this to be the other way around: L1 being all-sky and L2 only clear-sky.**

Right, this should have been the other way around. It is now corrected.

**lines 122-130 should be part of the Introduction, not results.**

This was re-organized, to some extent, and part of this text is now in the methods section.

**line 145 and figures 5 to 14.  There are far too many figures here that do not add too much information. These figures should be merged into two Figures maximum.**

In the revised version there is less figures included in the actual text, and many of the plots from different locations are now placed in the Supplement to make manuscript more easy to read. Also merging is conducted.

**Line 299: We performed the analysis on a seasonal basis and found that consistently the highest cloud related 300 AOD enhancements occur in sites in East-Asia.. Please,  write concisely,  these kind of statements make the manuscript very hard to read: make an analysis on the basis of seasonality and find a region that stands out.**

This is clarified.

---

## Author Response (AR2)

Dear Dr. Stier,

We would like to first express our thanks to you for your constructive and very useful comments. These comments below are in bold with our responses following after each comment. There is additionally a revised version, where we indicate by red color the changes and clarifications.

**Major issues:**
**1. The description of the used retrieval algorithms remains vague but the introduction seems to suggest that the used technique relies on direct-sun measurements, which are only possible for "thin enough" clouds. This is too vague. For any conclusions to be meaningful it needs to be entirely clear and reproducible what subset of observations the "cloudy sky" datasets represents. This also relates to comment 2 by reviewer 1, which has only been partially addressed. The conclusions refer to AOD enhancements of up to about 0.1 and suggest to include those in estimates of corresponding radiative effects but it is impossible to assess which fraction of the overall data this sample represents and what the corresponding synoptic / cloud regimes are. Some of the technical details are given in 2.1 but those do not allow to assess the overall impact on the representativeness of the retained retrievals.**

Thank you very much for these essential comments. We tried to explain these points in the first revision and apparently did not do a very good job. So we can certainly blame only ourselves that the same kind of miss-understandings are still possible. In this new revised version we tried to make these points clearer, hopefully sufficiently articulate and understandable. First, we do not carry out any retrievals of our own, but use publicly available SDA product and explain in detail the additional quality criteria we applied; so our results are easily reproducible, if one selects the data sets from AERONET as we have described. This algorithm is described in several papers and those are cited. The data have been also used before for similar studies and those are cited as well. The reader can easily re-produce the analysis and can moreover easily find additional information about the data and about the similar use of these data in earlier studies. To summarize very briefly: 1) the data are from AERONET SDA product, which is based on direct-sun measurements, 2) we select both L1 and L2 versions, latter including all the measurements and the former including only clear-sky measurements based on AERONET cloud-screening, 3) We also created a so called L0 data set, including measurements from L1 that are not in L2, and these "L0 measurements" are the "cloudy-sky" data set that you referred to above in your comment. So anyone can easily produce the exact same data set. The number of the L0 vs. L2 is illustrated by a histogram in the Supplement, so the fraction of these measurements is now given and stressed hopefully more clear in the revised version. Additionally, the "cloudy-sky" data is really primarily direct sun observations in the near vicinity of clouds, not mainly the observation of aerosol through clouds. See Eck et al. (2014) for detailed analysis of the AERONET cloud screening of these high temporal variance AOD observations near to clouds.

**2. With increasing cloud optical depth, the contribution of above cloud aerosol to the signal will presumably be attenuated, while below cloud aerosol still contributes. What role does the vertical distribution play? How sensitive are the results to the assumed cut-off of cloud optical depth? If the overall enhancement AOD under cloudy conditions is predominantly driven by boundary layer relative humidity then the impact on TOA direct radiative effects will be marginal so the meteorological regime and aerosol vertical distribution will matter.**

First of all, we did not explicitly assume any cut-off of cloud optical depth (OD) of our own. We use direct sun measurements in our analysis and sunphotometry can only measure max OD of ~7 for overhead sun and OD*m < 7 (where m=optical airmass) for increasing solar zenith angle. This is a basic upper AOD limitation of sunphotometer measurements in general, not just for AERONET, since the direct beam signal contribution nearly vanishes at this upper limit. Therefore the total OD (AOD + cloud OD) that can be monitored in cloudy conditions is limited to thin to thick cirrus, while the other cloud types have typically larger optical depths. These other cloud types typically have high enough temporal variability (in 15 seconds and 15 minutes), so that the triplet variance is too high for the measurements to pass to L2 (clear-sky) and it is often too high that they would be even included in L1 (RMS of raw counts >16%). Since Cirrus is typically at an altitude higher than the aerosol layer there is usually no enhancement of AOD with such cirrus observations.
Most of the enhanced fine mode AOD in L1 that are filtered in L2, which are the main interest in our study, are due to high triplet variability of fine AOD in proximity to cumulus clouds (Eck et al., 2014). And for these low altitude cloud bases there are multiple reasons for AOD to be enhanced near to clouds such as hygroscopic growth in high RH, gas-to-particle conversion, cloud processing of aerosol, convergence of aerosol associated with low pressure in cloud systems, etc. Also, for a limited set of observations of particular conditions, which were analyzed more carefully with the help of numerical modeling, it is obvious that set of observations is made through the cloud as droplet formation and scavenging of particles from the fine mode is needed to explain the observed change in AE.

Although the impact of enhanced AOD is relatively small to TOA fluxes, it was still warranted to mention it as being more representative to describe the columnar radiative effects. Because of strict cloud-screening used for L2 data, one would miss the cases of fractional cloudiness from the data and thus underestimate the AOD.

**3. The discussion of the AE enhancement through parcel modelling in 3.2 is interesting but I was missing a serious consideration of measurement and sampling uncertainties. The derived AE enhancements are small, but the uncertainties in fine mode AE retrievals (which is on its own highly derived product) are presumably not. Similarly, sampling errors may easily exceed these effects.**

We agree that the derived AE enhancements are small and that the AE uncertainty is higher than AOD uncertainty. However, we disagree that the analysis and related physical reasoning were not supported also from the measurement point of view, due to the measurement uncertainty. We had these two arguments in the manuscript and repeat here. First, if the box-plot had a huge range of values and mean and median just "happened" to be in the positive side, we would not have certainly made this same reasoning. However, now the negative values are essentially missing for the months of interest, so we argue that there is a true signal shown by the measurements. Second, we had these data as "AOD-weighted", so the most uncertain cases got less weight than the more certain AE cases of higher AOD.

**Overall, I was wondering if a demonstration of the retrieval accuracy under cloudy conditions based on synthetic data, allowing free variation of aerosol and cloud layers, would be required to demonstrate the accuracy of the approach. I may have missed if this has been done in some of the cited references, if so, some of this should be made more explicit in the introduction.**

The paper of Eck et al. (2014) has shown that the AOD enhancements in the vicinity of cumulus clouds that were measured by AERONET direct sun measurements were also measured independently by lidar instruments (both surface and aircraft based) and by in situ measurements of aerosol properties from aircraft profiles in the vicinity of these clouds. These enhanced AOD measurements near cumulus are typically removed by AERONET cloud screening since the algorithm operates primarily on temporal variance of optical depth (assuming cloud optical depth has higher variability) and the turbulent and dynamic environment near to clouds sometimes results in high frequency variation of AOD.

**Other issues:**

**4. Line 73: "We included all the good quality data available from all AERONET sites for global analysis." This is too vague.**

Instead of "good quality" we mentioned having selected the measurements from those time periods when Level 2 data are available which signifies good calibration and therefore high AOD accuracy, and also good instrument performance.

**5. Line 201: "The total column AOD and AE, as Cimel measurement would "see" from these modelled profiles". Avoid informal language. Cimel is not defined.**

This sentence has been changed.

**6. Fig. 3, 4: Are there other measurements available that could explain seasonality, e.g. RH?**

We consider that meteorological data analysis is beyond the scope of the present paper, but it is a good idea for a follow-up study.

[revised manuscript text omitted]